# Thermally Stable PVDF-HFP-Based Gel Polymer Electrolytes for High-Performance Lithium-Ion Batteries

**DOI:** 10.3390/nano12071056

**Published:** 2022-03-24

**Authors:** Devanadane Mouraliraman, Nitheesha Shaji, Sekar Praveen, Murugan Nanthagopal, Chang Won Ho, Murugesan Varun Karthik, Taehyung Kim, Chang Woo Lee

**Affiliations:** 1Department of Chemical Engineering (Integrated Engineering), College of Engineering, Kyung Hee University, Giheung, Yongin 17104, Korea; raman96@khu.ac.kr (D.M.); nitheesha@khu.ac.kr (N.S.); praveen.s@khu.ac.kr (S.P.); nanthamurugan@khu.ac.kr (M.N.); ghckddnjs@khu.ac.kr (C.W.H.); varun219@khu.ac.kr (M.V.K.); reotae@khu.ac.kr (T.K.); 2Center for the SMART Energy Platform, College of Engineering, Kyung Hee University, Giheung, Yongin 17104, Korea

**Keywords:** gel polymer electrolyte, solution casting, thermal stability, high safety, Li-ion batteries

## Abstract

The development of gel polymer electrolytes (GPEs) for lithium-ion batteries (LIBs) has paved the way to powering futuristic technological applications such as hybrid electric vehicles and portable electronic devices. Despite their multiple advantages, non-aqueous liquid electrolytes (LEs) possess certain drawbacks, such as plasticizers with flammable ethers and esters, electrochemical instability, and fluctuations in the active voltage scale, which limit the safety and working span of the batteries. However, these shortcomings can be rectified using GPEs, which result in the enhancement of functional properties such as thermal, chemical, and mechanical stability; electrolyte uptake; and ionic conductivity. Thus, we report on PVDF-HFP/PMMA/PVAc-based GPEs comprising poly(vinylidene fluoride-co-hexafluoropropylene) (PVDF-co-HFP) and poly(methyl methacrylate) (PMMA) host polymers and poly(vinyl acetate) (PVAc) as a guest polymer. A physicochemical characterization of the polymer membrane with GPE was conducted, and the electrochemical performance of the NCM811/Li half-cell with GPE was evaluated. The GPE exhibited an ionic conductivity of 4.24 × 10^−4^ S cm^−1^, and the NCM811/Li half-cell with GPE delivered an initial specific discharge capacity of 204 mAh g^−1^ at a current rate of 0.1 C. The cells exhibited excellent cyclic performance with 88% capacity retention after 50 cycles. Thus, this study presents a promising strategy for maintaining capacity retention, safety, and stable cyclic performance in rechargeable LIBs.

## 1. Introduction

In recent years, lithium-ion batteries (LIBs) have become the most extensively utilized portable energy storage devices owing to their numerous advantages, including low weight, low cost, high energy density, zero memory effect, extended cycle life, and maximum count of charge/discharge loops [1,2]. However, their shortcomings may have dire consequences in regard to safety issues, such as the leakage of liquid electrolytes (LEs), thermal instability, short-circuiting, and reliability associated with LIBs [3,4,5]. The widely used LEs of LIBs comprise a combination of lithium salts and organic carbonate solvents, which maintain good contact with the electrode and have high ionic conductivity and excellent rate capabilities [6,7]. Despite this, LEs also possess certain limitations, such as a greater chance of leakage and higher flammability. Meanwhile overheating of the battery can result in the emission of gases such as CH_4_, C_2_H_6_, and C_3_H_8_, historically resulting in catastrophic thermal accidents [8,9]. Hence, there is considerable demand for the development of a more reliable and safer electrolyte for extensive applications, without sacrificing the electrochemical properties of LIBs [10,11].

In the search for a highly safe and reliable electrolyte for LIBs, solid polymer electrolytes (SPEs) have gained significant scrutiny owing to their excellent features, including their free-standing structure, versatility, high flexibility, safety, and reliability [12,13]. Although safety issues and reliability are much improved, the use of SPEs for a wide range of applications is limited because of the poor electrode wettability and low ionic conductivity (10 − 8 × 10^−5^ S cm^−1^) [14,15]. To enhance the ionic conductivity of polymer electrolytes, the most effective way is to use low-molecular-weight carbonate solvents with lithium salts to develop a gel-type polymer electrolyte (GPE). GPEs can also act as a separator by serving as a physical barrier between the electrodes, exhibiting the cohesive and diffusive properties of solid and LEs [16,17,18]. Similar to LEs, GPEs exhibit excellent ionic conductivity and electrochemical stability [19,20]. GPEs are usually prepared by immersing a porous polymer membrane in an LE. However, the intrinsic safety issues caused by LEs can be avoided by using gel and solid-type electrolytes [21,22,23,24].

The GPE’s porous polymer membrane is prepared by dissolving the host and guest polymers in suitable solvents at an appropriate ratio. The emerging candidates for host polymers include poly(ethylene oxide), poly(vinyl chloride), polyacrylonitrile, poly(vinylidene fluoride)-co-hexafluoropropylene (PVDF-HFP), poly(vinyl alcohol) (PVA), and poly(methyl methacrylate) (PMMA). Among the polymers, PVDF-HFP appears to be the most promising polymer host owing to its excellent properties, such as semi-crystallinity, wider electrochemical stability, and excellent thermal and mechanical stability. In addition, the low glass transition temperature (Tg = −62 °C) and high dielectric constant (ε = 8.4) serve as a pathway for a higher dissociation of lithium ions by providing multiple mobile ions to enter the conduction process [25]. The crystalline part (VDF unit) gives the mechanical strength of the polymer matrix, and the amorphous part (HFP unit) arrests the LE (LiPF_6_/LiCF_3_SO_3_ with different carbonate solvents), helping enhance ionic conductivity (1.8 × 10^−2^ to 5 × 10^−2^ S cm^−1^) [26]. PMMA containing 96% amorphous content is preferably added to increase the ionic movement. The polymer matrix with a high amorphous nature enables local relaxation and creates segmental motion of the polymer chain for faster ion transport [27].

Considerable effort has been made to investigate PVDF-HFP/PMMA (PM)-based GPEs. For instance, Sundaram et al. developed MPPBM by blending PMMA with PVDF-HFP to investigate the porosity of the resultant polymer membrane. The as-fabricated MPPBM was found to have better ionic conductivity than MPPBM created using the conventional blending method. The added advantages of the prepared membrane included higher ionic conductivity, good adhesion to an electrode, and good thermal stability at high temperatures [28]. Building on this work, Xiao et al. developed a novel “sandwiched” membrane for use in LIBs comprising a PMMA film sandwiched between two layers of electrospun PVDF-HFP fibrous films. As a result, a membrane with ionic conductivity of 1.93 × 10^−3^ S cm^−1^ and an improved electrochemical stability of approximately 4.5 V (vs. Li/Li⁺), promoting lower activation energy for ion transport, was developed [29]. Meanwhile, Gebreyesus et al. investigated a polymer blend electrolyte comprising PVDF-HFP and PMMA as the host polymer and LiClO₄ as the salt, using the solution casting method. The resultant polymer blend electrolyte was used in LIBs with an optimized concentration ratio of PVDF-HFP (75 wt.%), PMMA (25 wt.%), and LiClO₄ [30]. Apart from PM-based electrolytes, PVAc/PMMA-based electrolytes have been widely studied because of their lower glass transition temperature, which enhances the conductivity [30,31,32]. In addition to conductivity, PVAc offers better wettability, promoting higher resistance against dissolution in non-aqueous electrolytes. For instance, Rajendran et al. developed a PVAc/PMMA-based electrolyte using the casting method, achieving an ionic conductivity of approximately 1.67 × 10^−4^ S cm^−1^ with excellent thermal stability [33]. Meanwhile, Baskaran et al. followed casting procedures to create a PVAc–PMMA–LiClO₄ polymer blend electrolyte using different salt concentrations. The polymer blend electrolyte containing 20 wt.% LiClO₄ had a reported ionic conductivity of 1.76 × 10^−3^ S cm^−1^ and lower activation energy, and was thus found to be the best polymer electrolyte for battery applications [34]. Overall, the electrolyte comprising PVDF-HFP and PMMA boasts enhanced mechanical and chemical stability as well as wettability, but lags in terms of electrocapacitive outputs, such as lower cyclic stability and poor rate performance [35]. In contrast, using PVAc-PMMA polymer blend electrolytes results in better electrochemical performance, but forfeits its mechanical stability [36,37,38,39]. Thus, to combine the advantages of both PM and PVAc–PMMA, PVAc was made with a blend of PM owing to its superior properties such as easy film formation, excellent mechanical strength, high solubility in LEs, and low glass transition temperature. In this study, using a simple and efficient solution casting method, PM, PVDF-HFP/PVAc (PC), and PVDF-HFP/PMMA/PVAc (PMC)-based highly flexible and porous polymer membranes were fabricated, and then immersed in 1 M LiPF_6_ in EC/DEC (1:1) LE to obtain highly rigid GPEs. The obtained GPEs exhibited excellent electrochemical and thermal stability and high lithium-ion conductivities. The electrochemical performance was analyzed using a CR2032 coin-type half-cell, which was assembled with LiNi_0.8_Co_0.1_Mn_0.1_O_2_ (NCM811) as the cathode, lithium metal as the anode, and PMC-based GPEs; the half cells exhibited excellent rate capability and cycling stability.

## 2. Materials and Methods

### 2.1. Materials Used

Commercial LiNi_0.8_Co_0.1_Mn_0.1_O_2_ (NCM811), which served as the cathode material, was purchased from L&F Co., Ltd. (Daegu, Korea), while the counter electrode lithium metal was obtained from Kanto Chemicals Co., Inc. (Tokyo, Japan) The commercial Celgard 2340 separator was purchased from Celgard Ltd. (Cheonan, Korea) The host polymer, PVDF-HFP, was purchased from Aldrich Chemicals, and the PMMA was purchased from Daejung Chemicals & Metals Co., Ltd. (Siheung, Korea). The guest polymer, PVAc, was purchased from Kanto Chemical Co., Inc. N, N-dimethylformamide (DMF) was purchased from Daejung Chemicals & Metals Co., Ltd., and acetone was purchased from Kanto Chemicals Co., Inc. (Tokyo, Japan) 

### 2.2. Preparation of PMC Polymer Membrane and Gel Polymer Electrolyte

The PMC polymer membrane was prepared using a simple solution casting method by varying the concentration of PVDF-HFP, PMMA, and PVAc, as shown in Table 1. PVDF-HFP was dissolved in an equal mixture of DMF and acetone at 40 °C for 2 h. Subsequently, the PMMA and PVAc polymers were dissolved in an equal mixture of DMF and acetone at 40 °C for 2 h under magnetic stirring. Both the polymer solutions were then mixed and stirred, resulting in a homogenous solution. The resultant polymer solution was transferred and cast onto a glass plate and dried at 60 °C for 12 h. As a result, a freestanding PMC polymer membrane was obtained, as shown in Figure 1. The optimized PMC polymer membrane comprised 60 wt.% PVDF-HFP, 20 wt.% PMMA, and 20 wt.% PVAc. To obtain the final Gelation phase and GPE, this dry polymer membrane was immersed in an LE (1 M LiPF_6_ in EC/DEC = 1:1, by volume ratio) for 4 h in an argon-filled glove box.

### 2.3. Physical and Chemical Characterizations

The morphology of the blended polymer membrane was analyzed using an FE-SEM analyzer LEO SUPRA 55 equipped with an EDAX spectrometer, Oxford Instruments (Abingdon, UK). The spectrometer (PerkinElmer, Waltham, MA, USA) was employed to record the FTIR spectra of the blended polymer membrane. The thermal properties of the blended polymer electrolytes were investigated via TGA analyzer, Q5000 IR, and DSC measurements were performed using an auto-modulated differential scanning calorimeter (Q-1000, TA Instruments, New Castle, DE, USA). The porosity of the membrane was determined by immersing the dry membrane in the LE, and expressed as follows:(1)$$Porosity\;\left(\%\right)=\frac{W_{wet}-W_{dry}}{(\rho_{ele})\left(V_{dry}\right)}\times 100\%\;$$
where *W_wet_* and *W_dry_* represent the weight of the polymer membrane after and before immersion in the electrolyte, respectively; *ρ_ele_* represents the density of electrolytes; and *V_dry_* corresponds to the apparent volume of the dry polymer membrane.

The electrolyte uptake and retention of the prepared polymer membrane were calculated by measuring the increase in weight of the membrane, as follows:(2)$$Electrolyte\;Uptake\;\left(\%\right)=\frac{W-W_{o}}{W_{o}}\times 100\%\;$$
(3)$$Electrolyte\;Retention\;\left(\%\right)=\frac{W-W_{o}}{W_{o}}\times 100\%\;$$
where *W_o_* indicates the weight of the polymer membrane before absorption of the electrolyte, and *W* indicates the weight of the polymer membrane after absorption of the electrolyte [40].

### 2.4. Preparation of Electrodes

The cathode electrode was prepared by the homogenous mixing of NCM811 as the active material, Denka acetylene black as the conductive agent, and PVDF as a binder in a mass ratio of 8:1:1. After grinding the materials, N-methyl-2-pyrrolidone solvent was added dropwise to form a uniform viscous slurry. The obtained slurry was coated onto battery-grade aluminum foil and dried at 110 °C for 5 h in a hot air oven. The dried cathode was punched into circular disks with a 14-mm diameter die for testing and kept in a vacuum oven at 100 °C for 5 h to eliminate moisture. The average mass loading of the cathode active material for all electrodes was approximately 5.3 mg cm^−2^ and the areal capacity of the electrode was found to be 1.082 mAh/cm^2^ while the thickness of the electrode was ~70 µm. The as-fabricated electrode served as a cathode for electrochemical characterization.

### 2.5. Electrochemical Characterization

All CR2032 coin-type cells were fabricated in an argon-filled glove box with controlled moisture and oxygen content, kept below 1 ppm. All electrochemical tests were conducted using an NCM811 as the cathode, lithium metal as the counter electrode, and the prepared GPE (PMC polymer membrane immersed in 1 M LiPF₆ (1:1 EC: DEC) electrolyte) as both the separator and electrolyte. The electrochemical cycling of the Li/GPE (PMC)/NCM811 half-cell was conducted at 0.1 C in a potential range of 3.0–4.3 V using an ETH cycler. All electrochemical measurements were performed at 25 °C. Linear sweep voltammetric (LSV) measurements were performed on SS/GPE/Li cells using an electrochemical analyzer (Iviumstat, Ivium Technologies) between 2.0 and 6.0 V at a scan rate of 1 mV s^−1^. Electrochemical impedance spectroscopy (EIS) was performed with an electrochemical analyzer in the frequency range of 100 kHz–0.01 Hz, with an amplitude of 5 mV. In addition, the ionic conductivity was estimated from the EIS data using SS/GPE/SS in EL cells.

## 3. Results and Discussion

The variations in the morphology of the PMC polymer membrane and PMC GPE were determined using FE-SEM analysis. Figure 2a,b show the morphology of the PMC polymer membrane. The PMC polymer membrane initially appeared to be rigid with PMMA and PVAc attached to it. Figure 2c,d show the morphology of the PMC GPE. After immersion into the LE for obtaining GPE, the PMMA constituted spherical particles as it dispersed in the solvent medium. The limited low solvency nature of PMMA is a major cause of the spherical particle morphology. The fibrous structure morphology seen in the FE-SEM images is the PVAc in the obtained GPE. In addition, PMMA seemed to be well incorporated into the PVDF-HFP and the contribution of the fibrous structure by PVAc polymer, providing the polymer matrix its robustness [41,42,43]. Moreover, this well-interconnected architecture provides porous nature to the GPE. The porous nature provides rich redox sites for Li-ion storage, thereby enhancing the electrochemical performance [44].

The FTIR technique is useful for determining the structural identity of polymers and blend polymers; the FT-IR spectra of the as-prepared GPE is shown in Figure 3a. The characteristic peaks at 834 and 870 cm^−1^ correspond to the amorphous phase of PVDF-HFP. The bands at 1073 cm^−1^ were assigned to the CH2 asymmetric stretching vibration of the PVAc [45]. The band around 1406 cm^−1^ was attributed to the bending vibration of CH_2_ in the PVAc, while the band located at 1728 cm^−1^ was assigned to the symmetric stretching of the carbonyl group of PMMA [45,46,47,48]. The observed peaks of PVDF-HFP, PMMA, and PVAc at their corresponding energies confirm the formation of the PMC polymer membrane.

The thermal transitions of the blended polymer membranes were investigated using DSC measurements. Figure 3b shows the DSC thermograms of PM, PC, and PMC. The DSC thermogram of PM shows a shift in the glass transition temperature (Tg) at 89 °C, which corresponds to the Tg of amorphous PMMA [46]. Additionally, the crystallization temperature of the membrane was found to be 141.27 °C, which occurred mainly due to the evolution of the gamma phase in the polymer. The thermogram of the PC complex showed endothermic transitions at 138.67 and 153.4 °C, ascribed to the Tg and melting temperature of the PVDF-HFP polymer. However, the PMC complex showed a shift in peak, exhibiting a crystallization temperature at 147.53 °C and a melting temperature at 160.87 °C. This implies an increase in the thermal stability of the blended polymer membrane, resulting from the blending of PVDF-HFP with PMMA and PVAc.

TGA was used to determine the thermal stability of each material and its fraction of volatile components in the GPE by monitoring the weight change that occurred when a sample was heated at a constant rate, as depicted in Figure 3c. The first decomposition temperature of the PMC membrane was found at 330–360 °C with a weight loss of 14.92%, which was caused by the thermal decomposition of PMMA and the deacetylation of PVAc. However, the thermal decomposition of PMMA resulted in the production of free radicals which, combined with successive depolymerization, occurred at higher temperatures [49,50]. The second decomposition temperature at 450–580 °C with a certain degree of weight loss may have been due to the thermal collapse of PVDF-HFP and polyene backbone degradation [45,51].

Thermal shrinkage behavior describes the ability of the polymer membrane to maintain its dimensions and integrity at higher temperatures, which is important for avoiding contact between positive and negative electrodes. To investigate the dimensional stability at elevated temperatures, the thermal shrinkage behaviors of the commercial Celgard separator, PM, PC, and PMC polymer membranes were recorded by placing the membrane on a hot plate at 100, 140, and 180 °C for 30 min, respectively. Figure 4 shows images of commercial Celgard separator, PM, PC, and PMC polymer membranes’ thermal shrinkage behavior. At 100 °C, Celgard, PC, and PM exhibited slight shrinkage, whereas PMC retained its dimensions. On increasing the temperature to 140 °C, the Celgard and PC membranes became transparent and shrunken. Further increasing the temperature to 180 °C caused Celgard and the PC membrane to completely lose their dimensions, while the PM membrane became brittle and started to deform. In contrast, the blended PMC polymer membrane deviated its dimensions slightly and retained a good degree of structural stability, proving that the membrane can withstand a higher temperature of 180 °C. Thus, it is evident that the combined effect of both the degree of cross-linking and rigid structure enhance the structural stability of the PMC polymer membrane. Meanwhile, the excellent/outstanding preservation of dimensional stability at high temperatures of the PMC GPE is enviable for sustainable energy storage devices at high temperatures [52].

The electrolyte uptake characteristics of the PM, PC and PMC polymer membranes were recorded by soaking the membranes in LE containing 1 M LiPF₆ in EC: DEC for 30 min [53]. The analysis of electrolyte uptake involves weight measurements of the polymer membrane before and after the absorption of the electrolyte. The electrolyte uptake was found to be approximately 200, 210, 340, and 679 wt.% for Celgard 2320, PM, PC and PMC polymer membranes, respectively (Table 2). Based on these results, it was concluded that the addition of PMMA and PVAc to the PVDF-HFP membrane caused an increase in electrolyte uptake in PMC polymer membranes, which attributed to the fibrous and spherical structures of PVAc and PMMA. This enhanced electrolyte uptake of PMC polymer membranes increases ionic conductivity, proving their suitability as separators for LIBs. Moreover, the PMC polymer membranes possess a higher electrolyte retention of 87.16% compared to those of the PM and PC. The PMC polymer membranes were completely wetted by the electrolyte due to their high porosity and pore volume, electrolyte-philicity of PVDF-HFP, and hydrophilicity of PMMA and PVAc. The electrolyte uptake and retention of the PM, PC, and PMC polymer electrolytes are shown in Table 2.

Furthermore, the flexibility and mechanical property of the PMC-based GPE was tested using physical methods. Figure 5 shows photographs of the PMC polymer membrane being subjected to flexibility tests at various positions before and after immersion in the LE for studying its preliminary mechanical properties. The PMC membrane was immersed in an LE containing 1 M LiPF₆ in EC: DEC for 30 min and a flexibility test was conducted. The PMC-based membranes before and after immersion demonstrated no signs of damage during various tests such as bending, rolling, and folding, confirming their excellent flexibility characteristics.

Figure 6a shows the ionic conductivities of PM, PC, and PMC at various temperatures using AC impedance spectroscopy. The PMC-based GPE was placed between stainless steel surfaces to fabricate SS/GPE/SS coin cells. The ionic conductivity can be calculated using:(4)$$\sigma =\frac{L}{A\;R_{b}}$$
where *L* and *A* are the thickness and area of the membrane, respectively, and *R_b_* is the bulk resistance from the EIS plot. The PMC exhibited a higher ionic conductivity of 4.24 × 10^−4^ S cm^−1^ compared to the ionic conductivities of PM (3.51 × 10^−4^ S cm^−1^) and PC (2.36 × 10^−4^ S cm^−1^). In addition, the ionic conductivity increased with increasing temperature, according to the Arrhenius equation:(5)$$\sigma =Aexp(\frac{-E_{a}}{KT})$$
where *A* denotes the pre-exponential factor; *K* and *T* are the Boltzmann constant and absolute temperature, respectively; and Ea denotes the activation energy. The PMC exhibited a lower activation energy of 0.60 eV compared to the activation energies of PM (1.70 eV) and PC (1.24 eV), promoting high lithium-ion transport to the PMC. With increasing temperature, the PMC membrane can expand and create a freer volume phase, resulting in increased polymer chain mobility, which in turn significantly enhances the transfer of ionic charge carriers. This appears to be the main factor contributing to higher ionic conduction in the PMC.

The LSV measurements of the PM, PC, and PMC GPEs were performed to determine their electrochemically active working potential stability window. Figure 6b shows the LSV curves of the PM, PC, and PMC in the voltage range of 2.0–6.0 V at a scan rate of 1 mV s^−1^. To function well, a good separator film must not undergo a redox reaction under the battery working conditions. As the energy density and voltage of the new generation of batteries increase, the separator film also faces the challenge of bearing a high voltage. The electrochemical window of 0–5.0 V is a new requirement for the separator film. As shown in Figure 6b, the oxidation potential of the PMC is 5.7 V, while that of the PM and PC are 5.2 and 5.3 V, respectively. This improved electrochemical voltage stability is attributed to the formation of a GPE, comprising a PMC separator film and LE. The gelation of the electrolyte makes it more difficult to decompose. For LIBs, better electrochemical stability implies a smaller polarization current during high-voltage charging and discharging, thus resulting in better cycling performance.

Figure 7a–c show the galvanostatic charge/discharge profiles of the NCM811/Li half-cell using PM-, PC-, and PMC-based GPEs. At a current rate of 0.1 C, the PMC’s initial specific discharge capacity was found to be around 204 mAh g^−1^, while the PM and PC delivered specific discharge capacities of 195 and 184 mAh g^−1^, respectively. The increased specific capacity of the PMC compared to that of the PM and PC was due to the synergetic effect of all three polymers. Figure 7d–f show the cycling performance and Coulombic efficiency of the PMC, PM, and PC at 0.1 C. For all three membranes, the initial specific charge capacity was high, resulting in low Coulombic efficiency during the initial cycles because of the irreversible electrochemical reaction. However, after some cycles, the Coulombic efficiency was maintained at 100%, demonstrating that the cycling performance was balanced and reversible.

Furthermore, a rate capability study was performed for the PM, PC, and PMC with current rates of 0.1, 0.2, 0.5, 1, and 2 C within the voltage window of 3.0–4.3 V, as shown in Figure 8a. A durable and stable rate capacity was observed at different charge/discharge rates for the PMC GPE membrane compared to the other two. The obtained reversible capacities were 204, 185, 167, 145, and 114 mAh g^−1^ at 0.1, 0.2, 0.5, 1, and 2 C, respectively, for the PMC GPE membrane. This was higher than those observed for the PM and PC. When the current rate decreased to 0.1 C, the specific discharge capacity of 183 mAh g^−1^ was recoverable and sustainable after 30 cycles without any major loss, indicating that the PMC GPE remained extremely stable during the extended rate cycling process. Such outstanding rate performance makes the porous polymer electrolyte membrane a favorable and promising candidate for high-performance LIBs. In addition, the rate capability of the PMC GPE was compared with that of a commercial Celgard separator using LE, as depicted in Figure 8b. The PMC GPE exhibited a discharge capacity of 204 mAh g^−1^, mainly owing to the activation of amorphous segments in the polymer membrane. The discharge capacity of LE was found to be 213 mAh g^−1^, which was slightly higher than that of the PMC GPE. The discharge capacity of LIBs depend on the cathode material employed; however, the electrolyte uptake and ionic conductivity of the electrolytes also influence the capacity of the battery. A higher percentage of LE uptake can wet electrode materials more sufficiently, helping to achieve high ionic conductivity for electrochemical reactions. Moreover, the unique structure of the PMC porous membrane can help to seal electrolytes and facilitate ion transportation, resulting in less capacity loss during the discharge and charge processes.

Figure 8c shows the cycling performance of the LE and GPE. From the graph, it is evident that the GPE exhibits almost equal capacity retention compared to that of the LE. For comparison, the GPE exhibited an initial capacity of approximately 204 mAh g^−1^. After 50 cycles, there was still a reversible capacity of 181 mAh g^−1^ remaining with a capacity retention of approximately 88%, while the LE with the Celgard separator retained a reversible capacity of approximately 190 mAh g^−1^ with a capacity retention of 89%. Thus, the prepared PMC (GPE) serves as an excellent electrolyte-cum-separator for LIBs, which is attributed to the combined effects of the PVDF-HFP, PMMA, and PVAc.

The results of the surface morphological analysis of the cycled GPE, bare NCM811 and cycled NCM811 electrodes are shown in Figure 9a–f. After the prolonged cycling of LIBs, the surface of the gel polymer electrolyte remains rougher with agglomerates and pores, which may be due to the transportation of lithium into the polymer membrane [6], as shown in Figure 9a. Figure 9c,d represent the morphology of the uncycled NCM 811 electrode with agglomerated structure. After cycling for 50 cycles, the NCM811 electrode appeared to decompose into smaller particles with a less porous structure, as shown in Figure 9e,f. The lower porosity is attributed to SEI formation arising from repeated cycling [7].

## 4. Conclusions

In this study, PMC GPE was prepared using the simple solution casting method. The combination of the three polymer matrices was found to improve the polymer membrane characteristics such as porosity, electrolyte uptake, and electrolyte retention. The as-prepared PMC GPE delivered an ionic conductivity of 4.24 × 10^−4^ S cm^−1^ with a lesser activation energy of 0.60 eV and wider electrochemical window of 5.7 V. Furthermore, owing to the remarkable compatibility of the PMC-based GPE with lithium metal and the improved oxidative stability with the cathode, the assembled NCM811/Li half-cell demonstrated an initial discharge specific capacity of 204 mAh g^−1^ at 0.1 C, with a capacity retention of 88% after 50 cycles and 100% Coulombic efficiency at room temperature, 25 °C. The improved porosity and thermal and electrochemical stability of the PMC GPE membrane make it a promising candidate for next-generation high-performance LIBs.

## Figures and Tables

**Figure 1 nanomaterials-12-01056-f001:**
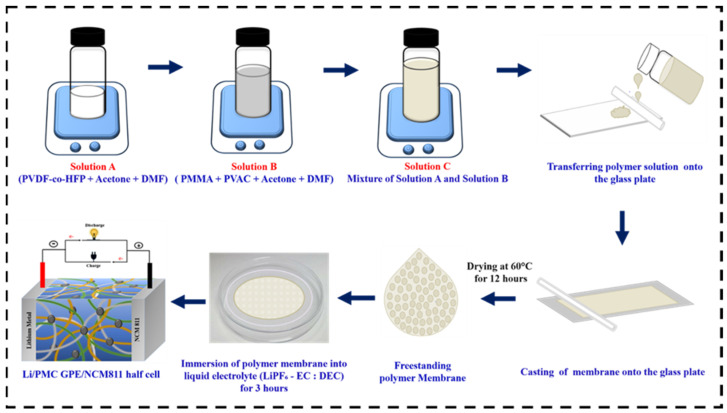
Schematic illustration of GPE synthesis.

**Figure 2 nanomaterials-12-01056-f002:**
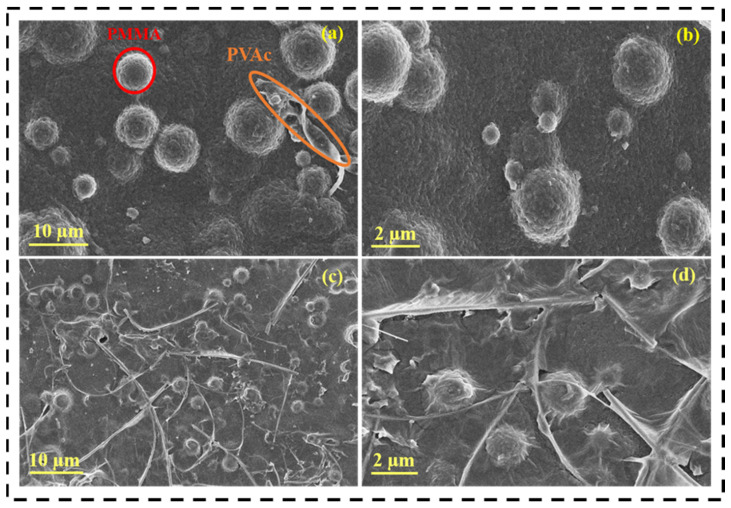
FE-SEM images of PMC polymer membrane (**a**,**b**) before immersion; (**c**,**d**) after immersion.

**Figure 3 nanomaterials-12-01056-f003:**
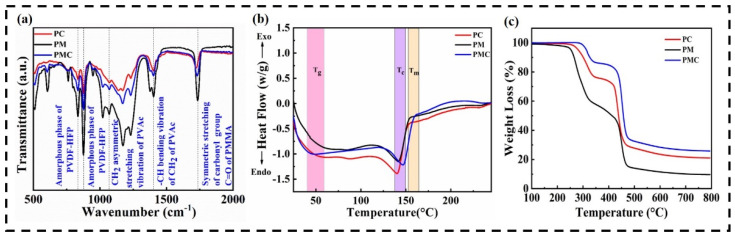
(**a**) FTIR spectra of PM, PC, PMC polymer membranes; (**b**) DSC curves of PM, PC and PMC polymer membranes; (**c**) TGA curves of PM, PC and PMC polymer membranes.

**Figure 4 nanomaterials-12-01056-f004:**
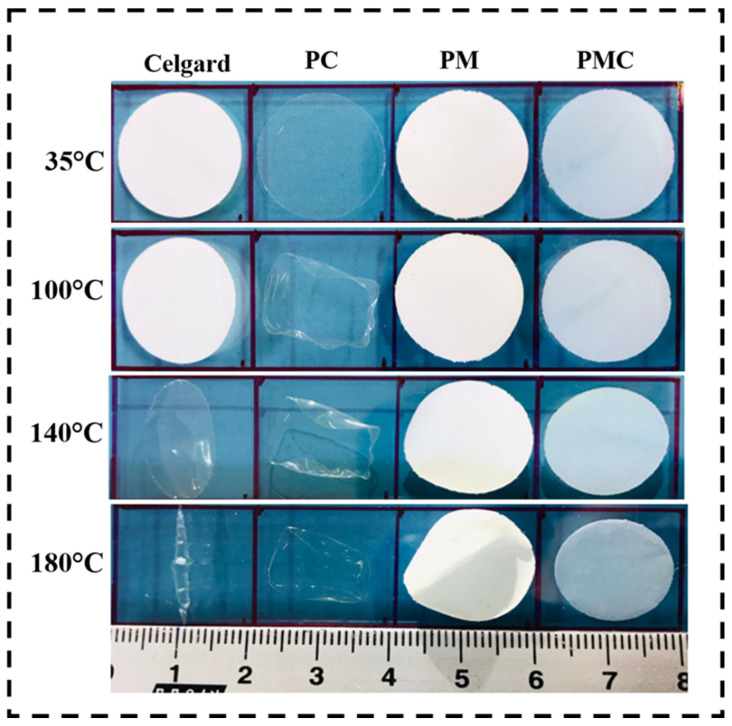
Photographs of polymer membranes subjected to a physical deformation test at various temperatures.

**Figure 5 nanomaterials-12-01056-f005:**
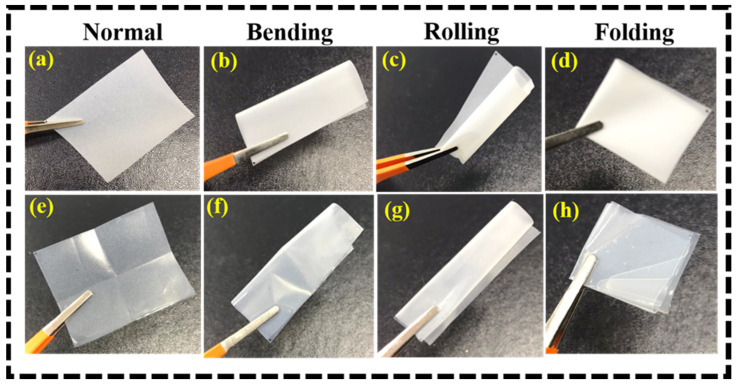
Photographs of PMC electrolyte membrane subjected to flexibility tests at various positions (**a**–**d**) before and (**e**–**h**) after immersion into the electrolyte.

**Figure 6 nanomaterials-12-01056-f006:**
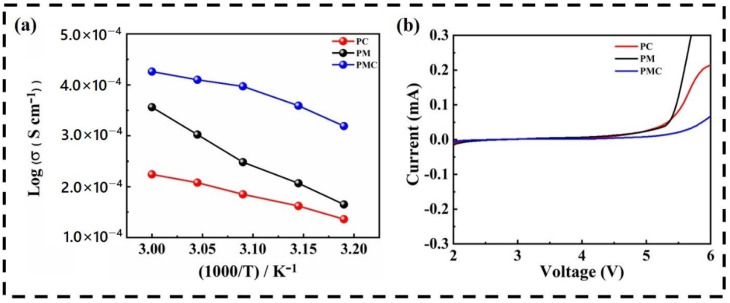
(**a**) Ionic conductivity measurements of PC, PM, PMC GPEs at various temperatures; (**b**) LSV plots of as-prepared dry polymer membranes.

**Figure 7 nanomaterials-12-01056-f007:**
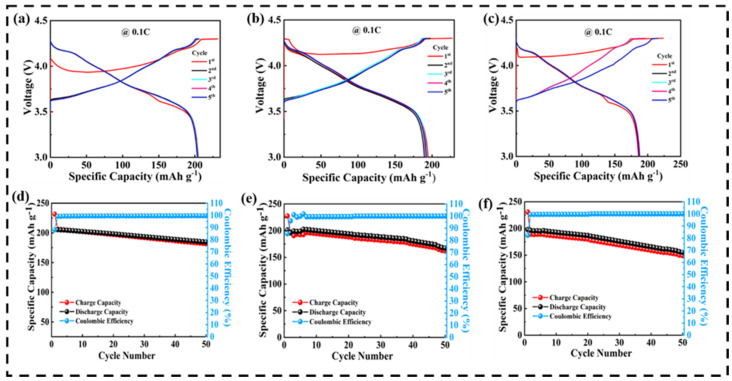
GCD profile of NCM811/Li half-cell using (**a**) PMC; (**b**) PM; and (**c**) PC; cyclic performance with Coulombic efficiency of NCM811/Li half-cell using (**d**) PMC-; (**e**) PM-; (**f**) PC-based GPEs.

**Figure 8 nanomaterials-12-01056-f008:**
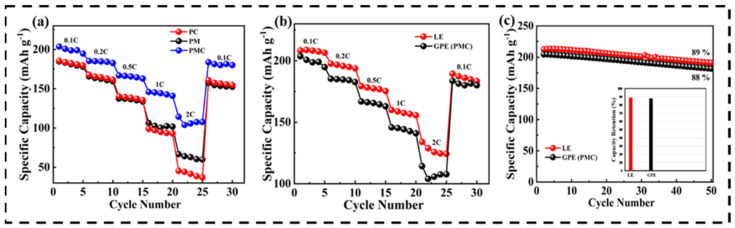
(**a**) Rate capability studies of NCM811/Li half-cell using PM-, PC-, and PMC-based GPEs; (**b**) Comparative rate capability studies for LE and GPE (PMC); (**c**) cyclic stability and capacity retention for LE and GPE (PMC).

**Figure 9 nanomaterials-12-01056-f009:**
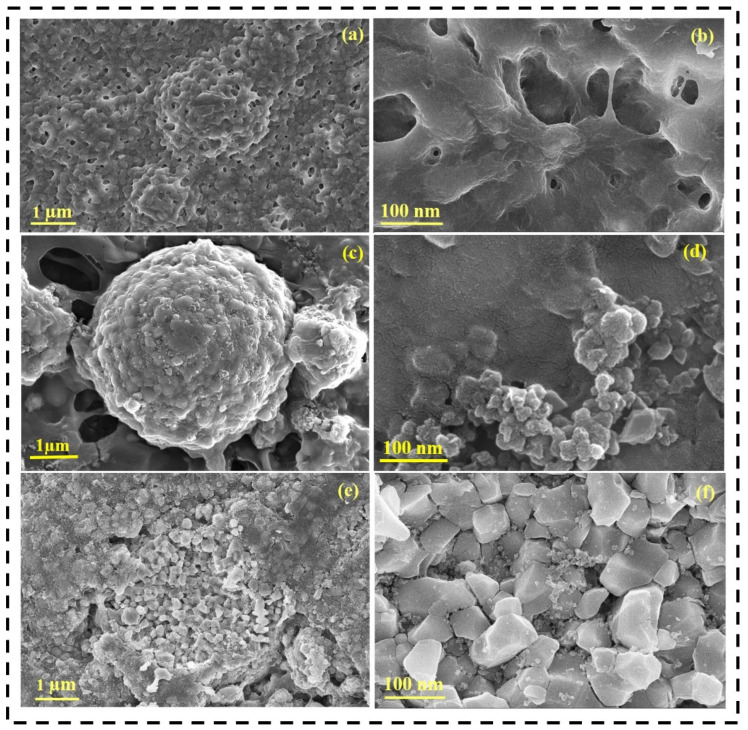
FE-SEM images of (**a**,**b**) cycled PMC GPE; (**c**,**d**) uncycled NCM811; and (**e**,**f**) cycled NCM811 electrode for 50 cycles at 0.1 C.

**Table 1 nanomaterials-12-01056-t001:** Composition of polymers used.

Sample Code	Used Polymers (wt.%)		
	PVDF-HFP	PMMA	PVAc
PM	60	40	0
PC	60	0	40
PMC	60	20	20

**Table 2 nanomaterials-12-01056-t002:** Comparison of electrolyte uptake, retention, and porosity of Celgard, PM, PC.

Sample Code	Electrolyte Uptake (%)	Electrolyte Retention (%)	Porosity (%)
Celgard 2340	200	86	38
PM	210	66.67	25.8
PC	340	77.27	39.4
PMC	679	87.16	47

## Data Availability

Not applicable.

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
