# Peer review of "Thermally Stable PVDF-HFP-Based Gel Polymer Electrolytes for High-Performance Lithium-Ion Batteries"

_nanomaterials, 2022, doi:10.3390/nano12071056_

Round 1

Reviewer 1 Report

The manuscript describe the preparation of polymer blends based on PVDF-HFP, PMMA and PVAc for LIBs applications. The research was well designed, with a plethora of experimental techniques. All the results support the authors’ conclusions. Consequently, this paper could be accepted for publication in Nanomaterials after some revisions.

In particular:

  • To elucidate the thermal properties of the blended polymer membrane, the authors presented DSC and TGA data. The characterization of thermal properties of the corresponding gels is completely missing. What is the effect of the liquid electrolyte on the thermal transition as well as on the thermal degradation of the gels?
  • Similarly, I cannot understand if the thermal shrinkage tests were carried out either on polymeric membranes or on GPEs. One can expect the presence of the LE strongly impacts on the flammability behaviour of the GPEs. Please clarify this crucial points and, in case, I suggest to perform the thermal shrinkage tests after swelling in LE.
  • I’m wondering if physical methods are enough to assess the mechanical performance and flexibility of the prepared GPEs. Would it be possible to perform DMA tests?
  • The highest conductivity of 0.4 mS cm-1 was achieved at 60 °C on PMC membrane. I suggest to compare the conductivity performance with similar systems such as PVdF-HFP comprising ionic liquids (Electrochimica Acta Volume 4011 January 2022 Article number 139470; DOI 10.1016/j.electacta.2021.139470)
  • The Activation Energy was calculated on three points. I’m wondering if they are enough to minimize the error on the calculation.
  • Please go through the manuscript to eliminate the typos

Reviewer 2 Report

The gel polymer electrolytes were developed and exhibited good performance for Li-ion batteries. which is interesting. It can be accepted for the publication after addressing the following comments:  

1.The good mechanical properties mentioned in the article are not supported by data. Authors need to provide the mechanical strength of the PMC polymer membrane.

2.Please check the oxidation potential of PMC in LSV and confirm whether it reaches 5.7 V.

3. In Fig. 3(b), please check the heat flow direction of the DSC curve.

4. The SEM images of the NCM811 electrode before cycle should be provided. 

Reviewer 3 Report

This manuscript reported on PVDF-HFP/PMMA/PVAc-based GPEs comprising poly(vinylidene fluoride-co-hexafluoropropylene) (PVDF-co-HFP) and poly(methyl methacrylate) (PMMA) host polymers and poly(vinyl acetate) (PVAc) as a guest polymer. In general, this work provides a new type of Gel Polymer Electrolyte for LIBs with high performance. However, there are several problems to be solved.

  1. Novelty should be established by referring similar papers published in the literature.
  2. English grammar needs a major revision.
  3. The experimental procedure is not clearly described. There are many unclear points, e.g., cathode loading level and the thickness.
  4. “The porous nature not only provide rich redox sites for Li-ion storage but also avoid stress/strain during charge-discharge process.” Avoid? Of course not!
  5. The annotation in Figure 3a is too small to see.
  6. The authors described the discharge/charge performance using C-rate. However, they have not provided any related information about the loading level, area specific capacity, thickness of the electrode. This does not make any scientific sense.

Round 2

Reviewer 1 Report

The revised paper is now clear, since experimental and discussion are now clearly exposed. To the opinion of the reviewer, the manuscript is worthily to be published in the present form.

Reviewer 2 Report

The authors have revised the manuscript as the comments, thus the paper can be published in Nanomaterials.

Reviewer 3 Report

The manuscript can be accepted in the present form.

However, a minor problem should be corrected. Line 175, "while the thickness of the electrode was Ì´70µ" should be "-70µm"